# Linking multiple serological assays to infer dengue virus infections from paired samples using mixture models

Marco Hamins-Puértolas[1]*, Darunee Buddhari[2], Henrik Salje[3], Angkana T. Huang[3], Taweewun Hunsawong[2], Derek A.T. Cummings[4,5], Stefan Fernandez[2], Aaron Farmer[6], Surachai Kaewhiran[7], Direk Khampaen[7], Anon Srikiatkhachorn[8,9], Sopon Iamsirithaworn[7], Adam Waickman[10], Stephen J. Thomas[10,11], Timothy Endy[10,12], Alan L. Rothman[8], Kathryn B. Anderson[10,11‡], Isabel Rodriguez-Barraquer[1,13‡]

1 Department of Medicine, University of California, San Francisco, California, United States of America, 2 Department of Virology, WRAIR-Armed Forces Research Institute of Medical Sciences, Bangkok, Thailand, 3 Department of Genetics, University of Cambridge, United Kingdom, 4 Department of Epidemiology, Johns Hopkins Bloomberg School of Public Health, Johns Hopkins University, Baltimore, Maryland, United States of America, 5 Department of Biomedical Engineering, Whiting School of Engineering, Johns Hopkins University, Baltimore, Maryland, United States of America, 6 Center for Infectious Diseases Research, Walter Reed Army Institute of Research, Silver Spring, Maryland, United States of America, 7 Ministry of Public Health, Tiwanond, Nonthaburi, Thailand, 8 Department of Cell and Molecular Biology, Institute for Immunology and Informatics, University of Rhode Island, Providence, Rhode Island, United States of America, 9 Faculty of Medicine, King Mongkut's Institute of Technology Ladkrabang, Bangkok, Thailand, 10 Department of Microbiology and Immunology, SUNY Upstate Medical University, Syracuse, New York, United States of America, 11 Upstate Global Health Institute, SUNY Upstate Medical University, Syracuse, New York, United States of America, 12 Coalition for Epidemic Preparedness Innovations (CEPI), Washington, D.C., United States of America, 13 Chan Zuckerberg Biohub, San Francisco, California, United States of America

‡ Indicates shared senior authorship
* marco.hamins-puertolas@ucsf.edu

## Abstract

Dengue virus (DENV) is an increasingly important human pathogen, with already half of the globe's population living in environments with transmission potential. Since many cases are missed by direct detection methods (RT-PCR or antigen tests), serological assays play an important role in the diagnostic process. However, individual assays can suffer from low sensitivity and specificity and interpreting results from multiple assays remains challenging, particularly because interpretations from multiple assays may differ, creating uncertainty over how to generate finalized interpretations. We develop a Bayesian mixture model that can jointly model data from multiple paired serological assays, to infer infection events. We first test the performance of our model using simulated data. We then apply our model to 677 pairs of acute and convalescent serum collected as a part of illness and household investigations across two longitudinal cohort studies in Kamphaeng Phet, Thailand, including data from 232 RT-PCR confirmed infections (gold standard). We compare the classification of the new model to prior standard interpretations that independently utilize information from either the hemagglutination inhibition assay (HAI) or the enzyme-linked

**Data availability statement:** All code written in support of this publication as well as simulation input files and generated data are publicly available at github.com/marcohamins/linking-multiple-sero-assays.

**Funding:** The authors were supported in this work by the following: NIH Grant P01 AI034533: entire team; Military Infectious Disease Research Program (MIDRP): DB, SF, AF, KBA; 1R01AI175941-01: entire team; R35 GM138361: MHP and IRB; European Research Council 804744: HS. IRB is a Chan Zuckerberg Biohub Investigator. Herchel Smith Postdoctoral Fellowship to ATH. The funders had no role in study design, data collection and analysis, decision to publish or preparation of the manuscript.

**Competing interests:** The authors have declared that no competing interests exist.

immunosorbent assay (EIA). We find that additional serological assays improve accuracy of infection detection for both simulated and real world data. Models incorporating paired IgG and IgM data as well as those incorporating IgG, IgM, and HAI data consistently have higher accuracy when using PCR confirmed infections as a gold standard (87–90% F1 scores, a combined metric of sensitivity and specificity) than currently implemented cut-point approaches (82–84% F1 scores). Our results provide a probabilistic framework through which multiple serological assays across different platforms can be leveraged across sequential serum samples to provide insight into whether individuals have recently experienced a DENV infection. These methods are applicable to other pathogen systems where multiple serological assays can be leveraged to quantify infection history.

## Author summary

Serological assays are crucial for diagnosing dengue virus infections, particularly when direct detection methods (RT-PCR or antigen tests) fail to identify cases. However, individual serological assays can suffer from sensitivity and specificity limitations that may lead to inconsistent results and classification difficulties. We developed a Bayesian mixture model framework that probabilistically infers infection events by integrating data from multiple paired serological assays. After validating our approach on simulated data, we applied the model to longitudinal cohort studies from Kamphaeng Phet, Thailand. Compared to standard interpretation methods, our model demonstrated improved infection detection accuracy as additional assays were incorporated in both simulated and real-world data applications. This probabilistic framework enables multiple serological assays to jointly provide enhanced insights into recent infection history. The methods are broadly applicable to any pathogen system where acute and convalescent data from one or more serological assays are available, offering a systematic approach for improving diagnostic accuracy through the integration of multiple assays.

## Introduction

Dengue virus (DENV) is a flavivirus spread by *Aedes* mosquitoes, primarily in tropical and sub-tropical regions of the globe. It is considered to be one of the leading causes of morbidity across the world [1,2]. With shifting geographical distributions of its vector driven by changes in climate, more populations will likely be at risk of infection in the coming decades [3–5]. A large proportion of infections are subclinical [6–8] meaning that estimates of infections can have large uncertainties when derived from surveillance of symptomatic cases alone. Diagnosis of dengue in clinical settings also remains challenging, since many cases are missed by direct detection methods (RT-PCR or antigen tests), and interpretation of serological assays may be

ambiguous if multiple assays disagree in their interpretations. Improved accuracy of diagnostic tools is important for multiple reasons; detecting missed infections, estimation of epidemiological parameters that provide insight into drivers of risk for both infection and disease, and when implementing and evaluating the efficacy of countermeasures like vector control or vaccine deployment [9–12].

Serological data is most commonly analyzed using defined cutoffs for both seropositivity and seroconversion [13,14]. However, these approaches may not be robust to variation in underlying population immunity and relative sampling times across populations that may lead to more misclassification [15]. In particular, differences in prior flavivirus exposure (e.g., vaccination programs, recent epidemics, original antigenic sin), sample timing [13], infecting serotype [16,17], underlying dengue strain utilized for antigen development in each assay, and whether the patient is undergoing a primary or post-primary infection [18] all could impact the sensitivity and specificity of this cutoff approach. Having low sensitivity due to false negatives may lead to underestimation of burden while low specificity due to false positives will lead to a misattribution of infection to dengue and could impact control efforts as well as vaccine deployment. In turn, optimizing diagnostic tools is vital in the management of dengue at both the population and individual patient scale. Mixture models have recently become more common in the field, allowing for data driven interpretations of serological data, [19–25] classification of DENV infections as primary or post-primary (secondary, tertiary, quaternary) [26,27] as well as to estimate the force of infection of DENV [26,28]. These methods are often highly generalized, making applications across different pathogens simple.

Similarly, paired sera (e.g., sera obtained at acute and convalescent time-points) are also usually evaluated using a cutoff metric to identify seroconversions or boosting events characteristic of infections. This cutoff is typically a 4-fold rise or greater for HAI, IgG, and IgM titers between the paired sera, [29,30] since it is expected that measurement error may lead to observed changes of up to a 2-fold rise. These cutoffs have been widely adopted across serological analyses [14,30,31] even though it is acknowledged that their performance may be affected by factors mentioned previously including the timing of sample collection and previous flavivirus exposure. Alternative statistical approaches have been developed to move away from these cutoff methods and have been successfully applied to paired sera from a single serological assay to define seroconversions [21,32]. However, these methods do not allow for the incorporation of multiple serological assays which can, in practice, increase the amount of available information through which estimates can be made. Using results from different serological assays has been shown to lead to tradeoffs in sensitivity and specificity [33–36], a balancing act characteristic of this classification problem. Recent development of multiplex serological panels only further demonstrates the need for methods that can leverage multiple data streams [37].

Here we develop a multivariate Gaussian mixture model approach for paired serological data (acute and convalescent samples) to probabilistically classify seroconversions from multiple serological assays. We test our method using simulated data, apply it to data from two longitudinal cohorts in Thailand, and compare it to existing approaches currently implemented in most DENV surveillance studies. These methods are flexible in the number of data streams that can be incorporated as well as allowing for the probability of infection to be assigned to an individual. Accurately diagnosing infections is a vital step in determining risk factors for infection and severe outcomes.

## Methods

### Ethics Statement

The protocol for KPS1 was approved by the Human Use Review and Regulatory Agency of the Office of the Army Surgeon General, the Institutional Review Board of the University of Massachusetts School of Medicine, and the Thai Ethical Review Board of the Ministry of Public Health, Thailand (protocol #654). The study protocol for KFCS was approved by Thailand Ministry of Public Health Ethical Research Committee, Siriraj Ethics Committee on Research Involving Human Subjects, Institutional Review Board for the Protection of Human Subjects State University of New York Upstate Medical University, and Walter Reed Army Institute of Research Institutional Review Board (protocol #2119). Formal written

consent was obtained from all adults and children were enrolled after both written assent and consent from the parent/guardian.

## Mixture model

Gaussian mixture models have successfully been applied to individual antibody titers in multiple pathogens including DENV and can provide an unbiased and unsupervised approach to quantifying seropositivity in a population of interest [19–23,28]. These methods implement a hierarchical probabilistic clustering framework where the data generating process is assumed to be produced in some predefined manner while associated parameters are subsequently estimated. We extend previous methodologies that either investigated multiple serological assays at a single time point, or a single assay at multiple time points to allow for the analysis of multiple serological assays measured at an acute and convalescent sample. The objective being to determine whether the patient is or is not acutely infected with the pathogen of interest, in this case DENV. In Fig 1 a set of theoretical antibody trajectories for a single assay are presented for infected (red) and uninfected (blue) individuals. Given an acute and convalescent pair of samples we developed a method, implemented in a Bayesian framework using stan, that can disentangle these two groups of infected and uninfected individuals. Here we will outline the methods used to infer infections from individual samples from a single assay, paired samples from a single assay, and then how to incorporate multiple assays taken at paired samples.

## Mixture models for serological data from a single acute or convalescent sample

We fit univariate mixture models to acute or convalescent samples independently (acute and convalescent geometric mean HAI, IgG, and IgM). We call these the acute or convalescent (GM HAI, IgG, or IgM) univariate mixture model hereafter. This is done using a single univariate mixture model with $M$ total mixtures, each of which has a mean and variance of $\mu_m$ and $\sigma_m$ respectively for each mixture $m$. The mixture with the highest average titers contained individuals who had the most recent boost in titers. These models were implemented to demonstrate the performance of each serological assay when only acute or convalescent samples are available. When fitting these models we choose the number of mixtures to be two, one for the infected and one for the uninfected classes.

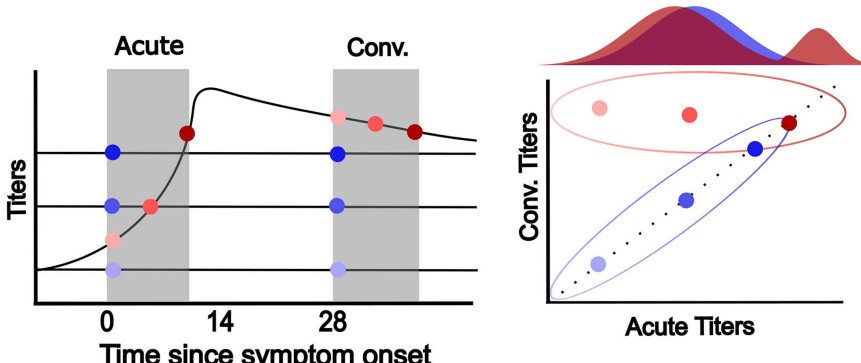

**Fig 1. Theoretical relationship between time since symptom onset and titers in one infected (red) and three uninfected (blue) individuals.** These titers are mapped onto a two-dimensional representation of acute and convalescent titers. Ellipsoids represent the expected distribution of acute and convalescent titers for infected and uninfected individuals while the underlying marginal distribution of these acute titers is presented at the top. Opacity of each pair of points represents titers at each acute sample. For the infected individual (red) the lightest point represents how titers compare when sampling close to the date of symptom onset while the darkest red point represents how titers compare when sampling more than a week from symptom onset, often defined as a "recent" infection. The marginal distribution of acute titers colored by infected and uninfected individuals is presented above the acute and convalescent titer plot to demonstrate how these recent infections can present.

## Mixture models for serological data from paired acute and convalescent samples

For paired serological samples we will first outline how this model is derived under the assumption that acute and convalescent samples were analyzed with a single assay. For each individual, $i$, in the sampled population of size $n$, we have measured acute and convalescent titers ($y_i^A$ and $y_i^C$ respectively). Let us assume that the likelihood function for convalescent titers is dependent on two normal distributions, one representing the distribution of convalescent titers given acute titers if an infection occurs and one when no infection occurs. We write this as follows:

$$f\left(y_i^C\right) = \theta_0 Normal\left(y_i^C | y_i^A, \sigma_0\right) + \theta_1 Normal\left(y_i^C | \mu + \beta\, y_i^A, \sigma_1\right),$$
(1)

Where $\theta_0 + \theta_1 = 1$, and $\theta_0$ represents the probability of not being infected while $\theta_1$ represents the probability of being infected. Let $\mu$ represent the minimum baseline boost of serological titers and $\beta$ be the relative increase in convalescent titers as a function of acute serological titers. In addition, $\sigma_0$ and $\sigma_1$ represent the variance of each normal distribution around their respective means.

For the multivariate case, where more than one assay's titers are incorporated, we extend the above to a multivariate normal distribution (MVN). We assume that each individual has acute and convalescent samples for k total assays measured as $y_{ij}^A$ and $y_{ij}^C$, where $j$ represents the assay number. Let $\hat{y}_i^A = \left(y_{i1}^A, y_{i2}^A, ..., y_{ik}^A\right)^T$ and $\hat{y}_i^C = \left(y_{i1}^C, y_{i2}^C, ..., y_{ik}^C\right)^T$ be the vectors containing all the observed acute and convalescent serological responses for an individual across all assays. These will contribute to the likelihood via two mixtures that are assumed to be multivariate normally distributed:

$$f\left(\hat{y}_i^C\right) = \theta_0 MVN\left(\hat{y}_i^A, \Sigma_0\right) + \theta_1 MVN\left(\hat{\mu} + \hat{\beta} \circ \hat{y}_i^A, \Sigma_1\right)$$
(2)

Again, let $\theta_0$ be the probability of not being infected while $\theta_1$ is the probability of being infected such that $\theta_0 + \theta_1 = 1$. Let $\hat{\mu}$ and $\beta$ now be vectors the length of the number of assays sampled ($k$) that represent the minimum baseline boosted serological titers and the relative increase in convalescent titers as a function of acute serological titers. Finally, let $\Sigma_0$ and $\Sigma_1$ be the covariance matrices associated with each mixture.

## Combining univariate and paired mixture models to identify recent infections

Since mixtures fit to paired acute/convalescent data may fail to capture infections where the acute titer was already very high (as illustrated in dark red in Fig 1), we combine univariate mixtures from acute samples with the paired mixture described above. This approach utilized the same mixture model approach outlined above to ensure differentiation of the high acute titers. For the identification of these recent infections we used a stepwise approach to combine an acute univariate model with a paired multivariate model. For the univariate approach we identified the number of mixtures that best explained the distribution of acute titers and assumed the mixture with the highest mean titers represented a recent infection. We determined if an infection can be considered recent by identifying if the assigned probability to this mixture with the highest mean titers was greater than 50%. For those that were not recent we found their probability of infection using the multivariate mixture model. We attempted to fit alternative approaches that incorporated the time since symptom onset as a covariate in the mixture but found these did not perform as well as this stepwise approach as multiple recent infections were missed (S1 Fig).

## Simulation of serological data

To assess the performance of our proposed model we first used simulated data generated using serosim [38], a flexible R package capable of simulating serological data which we ran in R version 4.3.3 [39]. We simulated the serological response to three serological assays (IgG, IgM, and GM HAI) for 500 individuals (Fig 2). This was done by defining exposure, infection given an exposure, and subsequent antibody kinetics post infection. The population as a whole underwent an extended

PLOS Computational Biology

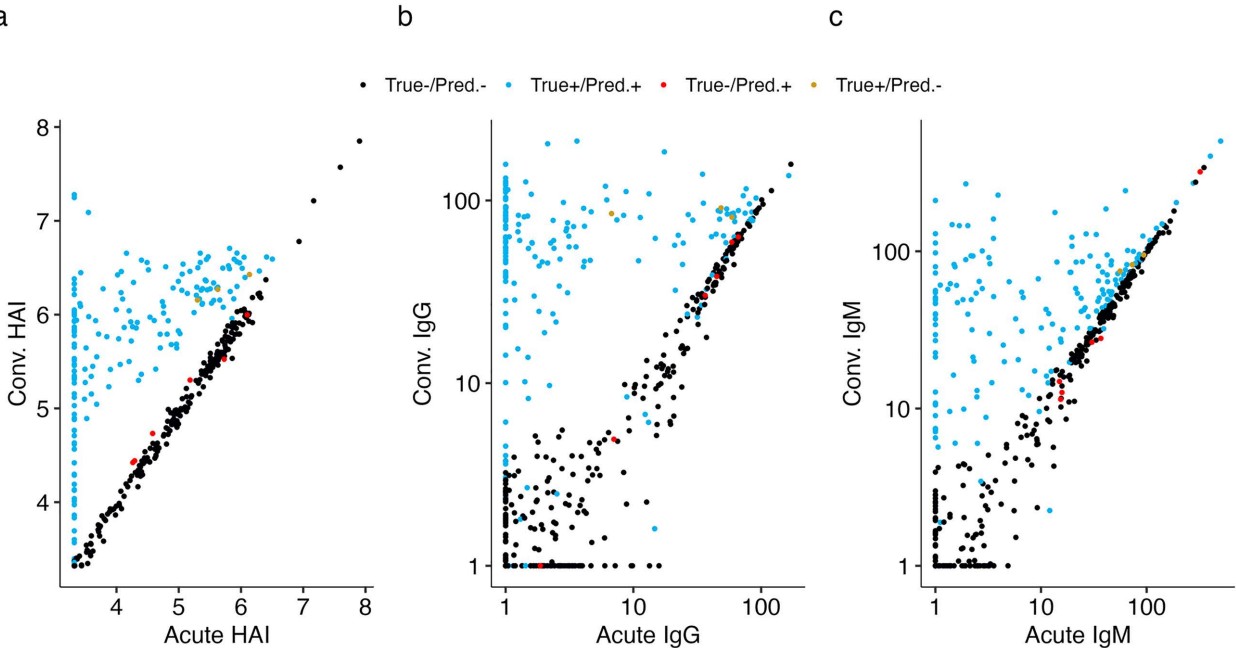

**Fig 2. Simulated serological data using serosim and subsequent results using the full mixture model on (a) HAI, (b) IgG and (c) IgM data with additional observational noise.** For each we present the predicted infection status using the mixture model approach along with the true underlying infection status. Parameters chosen for forward simulation are presented in S1 Table and resulting metrics of accuracy are presented in S4 Table.

period of low force of infection to the pathogen during which individuals could be infected, followed by a short period of high force of infection that infected approximately half of the population. After every successful infection an individual's antibody responses would undergo a boost in titers for all three assays followed by antibody decay. Upper and lower bounds of detection for each assay were pre-defined, and the relationship between an individual's initial titers and the expected boost in titers was also defined such that individuals with high titers experienced a smaller boost in titers. The chosen parameters for all antibody kinetics can be found in S1 Table. Note that this approach assumes an immediate boost post infection followed by a decay in titers, and does not consider differences in the timing of boosts that may exist between primary and post-primary infections. Simulated individuals could experience up to four sequential infections. Each individual's antibody profile, reflective of the past infection history and measured through antibody levels, impacted their probability of infection during a subsequent exposure window through a logistic transformation reflecting immune-mediated protection. Individuals with a simulated log2 GMHAI titer greater than 8 had a 50% chance of getting infected as compared to those with no titers. To generate the dataset for analysis, antibody titers consistent with an acute and convalescent time point were extracted from the simulated data. Three different amounts of observational noise were added to the data to quantify how measurement error impacts model performance. The relative observational noise used is presented in S2 Table allowing for direct comparison across assays as well as across the different datasets (simulated and cohort data). The code we utilized for these simulations can be found in the GitHub repository github.com/marcohamins/linking-multiple-sero-assays.

## Cohort data description

Data for this analysis was collected from two cohort studies. Both studies were conducted in Kamphaeng Phet province in Northern Thailand and their characteristics are presented in Table 1. The first cohort study, the Kamphaeng Phet Study (KPS1), recruited 3,647 children (aged seven to 15) from 12 different primary schools and ran from 1998 to 2002 [40,41]. The study involved blood draws four times a year (January, June, August, and November) as well

PLOS Computational Biology

**Table 1. Characteristics for Kamphaeng Phet cohort studies split by source (KPS1 and KFCS) as well as across the entire dataset. Information on age, hospitalization as well as serological assay results for RT-PCR, haemagglutination inhibition assay (HAI), and enzyme immunoassay (EIA) are presented.**

| Covariate | | Source | | |
|---|---|---|---|---|
| | | School-based cohort (KPS1, n=202) | Family cohort (KFCS, n=475) | Total (n=677) |
| Age (years) | Mean (SD) | 10.1 (1.5) | 11.0 (13.7) | 10.7 (11.6) |
| | Median [Min, Max] | 10 [7,15] | 6 [0,91] | 9 [0,91] |
| | [1,5) | 0 (0%) | 182 (38.3%) | 182 (26.9%) |
| | [5,18) | 202 (100%) | 206 (43.4%) | 408 (60.3%) |
| | [18,30) | 0 (0%) | 47 (9.9%) | 47 (7.0%) |
| | [30,100+) | 0 (0%) | 36 (7.6%) | 36 (5.3%) |
| Hospitalized Dengue | + | 40 (19.8%) | 48 (10.1%) | 88 (14.9%) |
| | − | 162 (81.2%) | 427 (89.9%) | 589 (85.1%) |
| RT-PCR | + | 157 (77.7%) | 75 (15.8%) | 232 (34.3%) |
| | − | 45 (22.3%) | 400 (84.2%) | 445 (65.7%) |
| HAI | Primary infection | 79 (39.1%) | 31 (6.5%) | 110 (16.2%) |
| | Post-primary infection | 109 (54.0%) | 64 (13.5%) | 173 (25.6%) |
| | No infection | 8 (3.9%) | 369 (77.7%) | 377 (55.7%) |
| | Recent infection | 6 (3.0%) | 11 (2.3%) | 17 (2.5%) |
| EIA | Primary infection | 8 (3.9%) | 16 (3.4%) | 24 (3.5%) |
| | Post-primary infection | 170 (84.2%) | 71 (14.9%) | 241 (35.6%) |
| | No infection | 23 (11.4%) | 378 (79.6%) | 401 (59.2%) |
| | JEV infection | 1 (0.5%) | 10 (2.1%) | 11 (1.6%) |

as active surveillance performed during the rainy season (June - November), where any children who experienced a febrile illness were evaluated and had acute and convalescent blood samples taken, with an interval of 14 days between collections. This led to 210 acute blood draws from active surveillance that underwent dengue RT-PCR while both acute and convalescent samples underwent serological testing (HAI, IgM, and IgG) for DENV and Japanese Encephalitis virus (JEV).

The second cohort study, the Kamphaeng Phet Family Cohort Study (KFCS) began in 2015 and remains active [42]. For this analysis we included all data collected between 2015 and 2022. During this period, 494 multigenerational households were enrolled consisting of 3220 individuals who provided yearly blood samples through routine visits. In addition, if an individual in the cohort reported a febrile event at any point during the year, an acute and two convalescent samples were obtained, with intervals of approximately 14 and 28 days. Only episodes where both the acute and first convalescent sample were collected were utilized in this analysis to ensure consistent sample timing across datasets (n=508). Similar to the previous cohort, acute samples underwent RT-PCR while acute and convalescent samples underwent HAI, IgM, and IgG testing for DENV and JEV. If this individual's acute sample was a confirmed DENV infection via RT-PCR, acute and convalescent specimens were collected from all other members of the household and underwent identical serological and molecular testing.

It is important to highlight that the data from these two cohorts were taken from two different subpopulations in Kamphaeng Phet and had different inclusion protocols. Individuals in KPS1 are those who had a confirmed DENV infection via RT-PCR, HAI, or EIA. However, in KFCS if someone in a household had a RT-PCR confirmed infection the rest of the household were also sampled in a cluster investigation. This can be found in the higher percentage of cases that were RT-PCR confirmed in KPS1 compared to KFCS.

## Laboratory methods and serological interpretations

A complete outline of the laboratory methods used for each serological assay are outlined in S1 Text and can be found in Hamins-Puertolas et al. (2023) [8]. When using HAI, infection is usually defined by a 4-fold or greater rise in HAI titers for any of the four DENV serotypes and JEV when comparing acute and convalescent sera. A higher seroconversion titer for JEV than for any DENV serotype is interpreted as a JEV infection event. All HAI titers we report are geometric mean HAI titers (GM HAI). For EIA units, an IgM ≥ 40 is usually used as a positive cutoff value. Evidence of DENV infection was classified by a DENV to JEV IgM ratio ≥ 1.0, and JEV infection when the ratio was < 1.0. Further classification into primary and post-primary for each assay is outlined in S1 Text.

## Model fitting

All models were implemented in stan [43]. We fit 15 total models: Models 1–9 incorporated data from each combination of the three assays at both acute and convalescent time points (Table 2) while Models 10–15 fit univariate mixture models to one assay and time point (acute or convalescent) at a time (S6 Table). Models 7 and 9 combined multivariate and univariate mixture models to account for high acute titers. All models were estimated using Cmdstanr v0.5.3 with four independent chains, each of length 1000 after 2000 discarded for warm-up and model code can be found in the github repository github.com/marcohamins/linking-multiple-sero-assays [44]. Convergence was defined to be when R-hat < 1.1 and the effective sample size >300 for model parameters, after which posterior draws were pooled for reported parameter estimates. To assess accuracy we performed 4-fold cross-validation with testing set predictions used to assess accuracy. Bayesian priors are outlined in S3 Table.

We removed any individuals whose GM HAI titers had reduced between acute and convalescent samples by more than 25%. These waning titers may reflect a DENV infection in the previous few months, either missed by the active surveillance program or subclinical. However, since no determination can be made, we removed these individuals from classification (n = 41) leading to a total of 677 total samples for which accuracy is assessed [45].

**Table 2. Comparison of each accuracy metric (sensitivity, specificity, F1, and area under the curve for reporter operator curve (AUC ROC)) on each model. Models are defined by which assays were incorporated (i.e., HAI, IgG, and/or IgM). Bolded values represent the best performing model for the infection definition and metric of interest.**

| Model | | Infection definition | | | | | | | |
|---|---|---|---|---|---|---|---|---|---|
| | | RT-PCR + vs. RT-PCR - | | | | RT-PCR, HAI, or EIA + vs. RT-PCR, HAI, and EIA - | | | |
| # | Data | Sens. | Spec. | F1 score | AUC ROC | Sens. | Spec | F1 score | AUC ROC |
| – | HAI clinical interp. | 98.3% | 82.5% | 83.7% | – | *94.8%* | *100%* | *97.4%* | – |
| – | EIA clinical interp. | 90.1% | 85.6% | 81.8% | – | *84.5%* | *100%* | *91.6%* | – |
| – | HAI AND EIA clinical interp. | 89.7% | 88.3% | 83.7% | – | *80.3%* | *100%* | *89.1%* | – |
| – | HAI OR EIA clinical interp. | 98.7% | 79.8% | 81.9% | – | *99.1%* | *100%* | *99.5%* | – |
| 1 | IgG | 90.5% | 93.5% | 89.2% | 95.9% | 72.4% | **100%** | 84.0% | 96.8% |
| 2 | IgM | 70.0% | 94.8% | 77.7% | 93.2% | 55.8% | 99.7% | 71.5% | 93.7% |
| 3 | GM HAI | **97.0%** | 80.8% | 83.0% | 92.9% | **90.3%** | 96.5% | **93.1%** | **98.4%** |
| 4 | GM HAI + IgM | 95.7% | 86.5% | 86.6% | 94.4% | 85.5% | 99.7% | 92.0% | 98.2% |
| 5 | GM HAI + IgG | 96.1% | **97.1%** | 86.9% | 94.4% | 85.5% | **100%** | 91.6% | 97.9% |
| 6 | IgM + IgG | 91.8% | 93.0% | **89.5%** | **96.1%** | 73.9% | **100%** | 85.0% | 96.4% |
| 7 | IgM + IgG + acute IgG mixture | 94.4% | 85.3% | 84.9% | 95.0% | 86.1% | **100%** | 92.5% | 96.6% |
| 8 | All | 96.1% | 86.7% | 86.8% | 94.4% | 85.2% | 99.7% | 91.8% | 97.6% |
| 9 | All + acute GM HAI mixture | 96.1% | 86.2% | 85.4% | 94.5% | 85.2% | 97.7% | 90.6% | 97.8% |

## Assessing model performance

In order to directly compare the results from each of the models against both the simulated and the cohort data, we utilized sensitivity, specificity, area under the reporter operator curves (ROC AUC), and F1-score. The F1-score is a measure of predictive performance that weights the number of correctly identified infections (TP) by the number of missed infections (FN) and incorrectly identified non-infection events (FP). The precise formula is TP/(TP + 0.5*(FP + FN)). For the Thai data we presented these evaluation metrics for two definitions of what a true positive infection was for classification purposes. The first evaluation metric only considered RT-PCR positive infections while the second definition considered an individual to be infected if either RT-PCR, HAI, or EIA were positive. The former being a more specific definition since RT-PCR will have few false positives and more false negatives while the latter is more sensitive due to allowing either RT-PCR or serological results to define an infection.

## Results

### Assessment of model performance to identify infections using simulated data

We first evaluated the performance of models on multiple sets of simulated data. We simulated paired (acute/convalescent) serological data for three independent biomarkers (GM HAIs, IgM and IgG) for 500 individuals of whom approximately 50% had been recently infected with a pathogen we intended to detect.

We first fit models incorporating data from the three simulated assays (GM HAIs, IgM and IgG) to try and infer the infection status of each individual (Fig 2). The model incorporating all three assays had an F1 score increasing from 93.9-98.0% as observational noise was decreased. This model consistently outperformed the models that incorporated any one or two of the assays when comparing sensitivity, F1, and AUC ROCs (S4 Table). Specificity was also highest in this model in all but one observational noise simulation. We then fit models including paired acute and convalescent samples from individual and combined sets of assays, leading to seven total models. Models including any two combined assays outperformed any singular assay, with increases in F1 scores and AUC ROC ranging from 0.1 to 27.3 and 0.1 to 8.0 percentage points respectively across the various observational noise simulations. The largest increases in performance were improvements on models that just had IgM or IgG data incorporated. The incorporation of multiple assays also reduced the decrease in performance that was experienced across all models as observational noise increased. For example, the model incorporating just HAI had F1 scores decrease 12.3% points as observational noise increased while models incorporating HAI and another assay (either IgG or IgM) had an average decrease in F1 score of 6.8%. Two models incorporating just IgM were unable to be fit at higher levels of observational noise, likely due to a loss of signal from titer boosts post infection. The remaining six models all performed well in scenarios ranging from small to large amounts of observational noise.

### Application of the model to paired serological data from Thailand

We then assessed the performance of our methods on data from the two longitudinal cohorts based in Kamphaeng Phet, Thailand, where 677 paired sera from 560 individuals were taken that had 232 RT-PCR confirmed infections. We assessed the performance of 15 models under two different definitions of infection, described in the Methods section. This allowed us to compare a more specific gold standard (infections confirmed by RT-PCR) where few false positives and more false negatives are expected, to one that is more sensitive (infections defined by RT-PCR or serological results). We ran a total of 15 models, each model incorporated a different number of samples (acute and/or convalescent) and assays (all combinations of IGM, IgG, and HAI). This includes the seven models tested on simulated data, as well as variations developed to better capture recent infections where serological titers are already high at the acute sample, as well as single time point mixture models for all assays at both acute and convalescent samples.

For the first definition, using RT-PCR results, we found that all models except one (Model 2 with just IgM data) performed as well if not better than the previously implemented classifications based solely on HAI (HAI interpretation), IgM

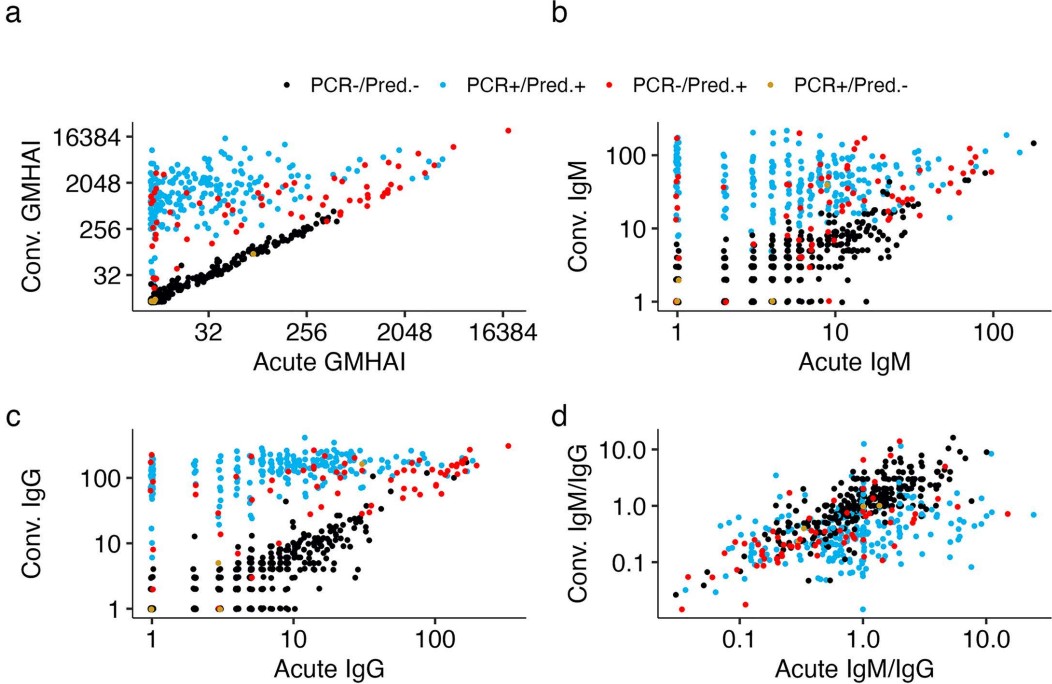

**Fig 3. Acute and convalescent serological data from Thai cohorts used in this analyses, colored by true infections status and prediction of infection during model testing. a)** Geometric mean titers of a haemagglutination inhibition assay (GM HAI) for all four serotypes of dengue virus. **b)** Immunoglobulin M (IgM) **c)** Immunoglobulin G (IgG) **d)** Ratio of IgM to IgG at both acute and convalescent sera samples.

and IgG values (EIA interpretation), or combinations of these interpretations (Table 2). For this definition we found that the best performing approach according to the F1 metric and AUC ROC was the model that incorporated information from just IgM and IgG data (Model 6). However, the mixture that combined all data sources had better sensitivity (96.1% in Model 9 vs 91.8% in Model 6), which may be a more relevant metric of performance for this highly specific definition. With the more sensitive definition of infection that incorporates RT-PCR, HAI, and EIA information we found that using just GM HAI data (Model 3) resulted in the highest F1 and AUC ROC scores. All models with two or more data streams except Model 6 (IgG and IgM) had high F1 and AUC ROC scores (>90%). Note that the comparisons to the HAI- and ELISA-based interpretations for this gold-standard definition of infection are not meaningful since the interpretation of infection is part of the gold-standard definition, however we reported these values for full transparency (i.e., specificity compared to this definition was 100% in all of these clinical interpretations by construction).

Overall, we found that combining multiple data streams using this mixture model approach consistently led to higher performance when compared to currently implemented interpretations. Model 9 (GM HAI, IgG, IgM, and an acute GM HAI univariate mixture model, Fig 3) consistently performed well across both RT-PCR and combined serological definitions of infection (Table 2), where it had high sensitivity (96.1% and 85.2%) and specificity (86.2% and 97.7%). We also found that when a DENV infection was classified by one, two or all three assays (RT-PCR, HAI, and EIA illustrated in S2 Fig), this model identified an infection in 48.7% (19/39), 97.0% (65/67), and 100% (208/208) of individuals respectively (S3 Fig, S5 Table). This model also identified three additional infections that were missed by all three interpretations, two of which corresponded to high acute and convalescent GM HAI titers while one was near the decision boundary for the IgM paired samples and had an inferred probability of infection of 57% (S4 Fig). In addition, Model 7 (IgM, IgG, and a separate acute IgG classifier), consistently performed as well or better than the currently implemented algorithms. In individuals where one, two, or all three serological assays (RT-PCR, HAI, and EIA) classify there to be an infection, this model correctly

identified an infection in 30.8% (12/39), 85.1% (57/67), and 97.1% (202/208) of individuals respectively with no additional infections inferred when all three serological assays did not classify there to be an infection.

In addition to performing equally or better than a standard algorithm, a useful feature of the proposed mixture model methodology is that beyond classifying individuals, it provides an estimate for the probability that an individual belongs to the infected class, and thus the "certainty" of classification (S5 Fig). In the aforementioned Models 7 and 9 we find that 85.5% (579/677) and 96.9% (656/677) respectively of paired sera have an inferred probability of infection below 5% or above 95%, indicating high certainty in a sizeable majority of cases. For the remaining paired samples with lower certainty (between 5 and 95% inferred probability of infection), Models 7 and 9 had 50% (49/98) and 67% (14/21) from disagreeing interpretations (i.e.,. one or two positive serological interpretations) respectively(S3 Fig).

Finally, we compared the performance of our paired acute/convalescent models to univariate mixture models independently fit on acute or convalescent data for each serological assay (S6 Table). We found that each acute data stream performed poorly against both definitions of infection. Titers had not sufficiently risen to distinguish infected individuals from uninfected individuals. The univariate convalescent mixture models performed better than their acute counterparts with increases in F1 and AUC ROC scores ranging from 19.8 to 56.7 and 39.1 to 48.1 percentage points. Convalescent IgG and IgM both performed at similarly high levels within these univariate models. For example, when using the RT-PCR definition of infection with F1 scores of 66.4% and 67.4% and AUC ROC scores of 92.6% and 91.3% while the convalescent GM HAI had scores of 56.8% and 90.9%. None of these models performed better than models that incorporated both acute and convalescent titers from the same assay demonstrating the importance of acute samples in providing background immunological context for individuals, but that reliance on acute samples alone is not sufficient for accurate diagnosis.

### Estimation of parameters related to antibody kinetics

We also investigated whether these methods can infer parameters associated with antibody kinetics. For the simulated data, we find that we are able to reconstruct the simulated relationship between acute titers and expected boosts across all three assays (S6 Fig). The reconstructed relationship remains consistent across varying levels of observational noise. Although estimates do diverge from the ground truth, this is due to the functional form constraint defined by the mixture models. We also estimate this same value for the Thai dataset and find that the expected GM HAI, IgG, and IgM additive titer boost in a fully susceptible individual are 160, 36, and 17 respectively (S7A). The relationship between this expected boost and an individual's acute titers is also quantified (S7B Fig).

### Discussion

We have developed a probabilistic framework through which paired acute and convalescent serological measurements from multiple assays can be jointly leveraged to predict the probability of an acute or recent infection. We first used simulated data to demonstrate the performance of these methods under varying levels of observational noise. We then applied these methods to real world serological data from Kamphaeng Phet, Thailand, and showed that we were able to accurately classify infections as well as quantify boosting patterns as a function of acute titers. In general the incorporation of additional data streams led to higher performance in both simulations and real world applications.

We validated the methodology using simulated paired serological data. We found that observational noise reduces performance, but that these reductions in performance can be minimized when additional data streams are incorporated. In these simulations the observational noise across assays was assumed to be statistically independent, meaning each additional assay provided a unique draw centered around the true serological response. In turn, sharing information across these assays should simultaneously minimize false positives and false negatives. Previous work has demonstrated that a simple combination of classification results can lead to increased sensitivity, often at the cost of reduced specificity [33–36]. We show that when applied to simulated data these methods are capable of increasing both sensitivity and specificity as additional immunological data is leveraged under increasing levels of observational noise.

When applied to data from Kamphaeng Phet, Thailand, mixture models are capable of probabilistically assigning whether paired acute and convalescent serological results (taken between 14–28 days apart) from multiple assays are consistent with an infection. These methods result in high performance when compared to the gold standard RT-PCR as well as combined serological interpretations, and can resolve discrepancies in conclusions drawn from RT-PCR, HAI, and EIA methods across a variety of assay combinations. Although these results are context dependent, we find that in Kamphaeng Phet mixture models based solely on the acute and convalescent GM HAI titers perform well in identifying infections and could be used alone in a resource limited setting for this purpose. However, in contexts where HAIs are not available we find that using just paired samples tested with EIA (IgG and IgM) provided some of the best available information from which interpretations could be made. Using acute samples alone leads to drops in classification accuracy while convalescent samples alone perform relatively well in this task even without the additional context that acute samples provide. This itself is not surprising as the majority of rises in titers will be found during the convalescent sample while acute data provides a baseline against which subsequent rises can be compared.

One of this study's primary strengths is the two cohorts of data that provide an ideal dataset on which to assess the performance of this methodology. By combining a cohort focused on individuals around the average age of infection (school aged children) and one that sampled individuals across a wide age range and immune profiles using multiple serological assays allowed for this flexible framework to be tested. Due to these study characteristics we were also able to show how methods like these can be implemented in different scenarios (varying number of assays) highlighting the flexibility of this approach. This allows for investigators to infer the probability of infection from paired acute and convalescent data in a semi-supervised manner. In addition, this provides a systematic way through which the predictive power of each assay and combinations of assays can be compared and in turn be used to inform future study design. However, this work has some limitations. One major limitation is that the paired model is at times not sufficient and two independent models (recent infection and paired model) had to be combined to explain paired samples with high acute titers (recent infections). We worked to include additional covariates into the models like baseline titers prior to the illness investigation and time since symptom onset to explain these initially high titers but found these models were still unable to fully capture recent infections. Future work should strive to leverage time since symptom onset as well as available immunological data that provide a baseline measure of immunity. In addition, although this approach was run using a training and testing approach to quantify accuracy in out-of-training samples and we used data from two independent studies, it would be useful to assess the performance of these methods in other populations with different underlying infection histories and immune profiles (i.e., how transferable is classification). Note that these methods have been applied to HAI and EIA data from in-house assays that are likely qualitatively consistent with other studies, but means these results cannot simply be exported and applied to other datasets. However, the applicability of this work to other studies both within the DENV literature as well as in different pathogen systems remains.

Optimizing the interpretation of serological data is critical for epidemiological and clinical studies of diseases like dengue, where the availability and sensitivity of direct detection methods may be limited. Here we present a mixture model based approach to probabilistically infer infection status from this paired serological data across multiple assays. This allows for the estimation of uncertainty, particularly useful when currently implemented interpretations may conflict. Beyond dengue virus and other arboviral pathogens, these methods are highly transferable and can be applied to any pathogens where paired data from one or more assays are available. As multiplex assays become more common in epidemiological studies, methods that leverage all available data simultaneously will become increasingly necessary. We have provided code for both the simulation approach as well as the full mixture model methods, both implemented in R, to stimulate further exploration of these methods when applied to serological data.

## Supporting information

**S1 Fig. Acute and convalescent data for serological assays, colored by probability of.** infection found during model testing for a model that accounts for time since symptom onset. a) Geometric mean titers of a haemagglutination inhibition

assay (GM HAI) for all four serotypes of dengue virus. b) Immunoglobulin G (IgG) c) Immunoglobulin M (IgM) d) Ratio of IgM to IgG at both acute and convalescent sera samples.
(TIF)

**S1 Table. Parameters for forward simulation using serosim [38].** Unless otherwise noted all parameters are drawn from log-normal distributions.
(XLSX)

**S2 Table. Relative observational noise by assay and dataset.** Methodology outlined in the *Relative observational noise estimation* section of S1 Text.
(XLSX)

**S1 Text. Description of serological assay methodologies, alternative mixture model methods, and estimation of relative observational noise.**
(DOCX)

**S3 Table. Parameters found in Bayesian estimation framework, definition, and prior distribution.**
(XLSX)

**S4 Table. Model comparisons for simulated data with differing amounts of observational noise.** Parameters associated with these simulations are presented in S1 Table. Models with convergence issues have an NA presented.
(XLSX)

**S5 Table. Distribution of paired serological data that fall into probability buckets when using Model 9 (incorporating paired IgG, IgM, GM HAI, and acute GM HAI serological data) and the number of serological assays (RT-PCR, HAI, and EIA) that interpret the individual as having an infection.**
(XLSX)

**S2 Fig. Acute and convalescent samples from cohort study data for serological assays, colored by the number of serological assays (RT-PCR, HAI, and EIA) that interpret the individual as having an infection. a) Geometric mean titers of a haemagglutination inhibition assay (GM HAI) for all four serotypes of dengue virus. b) Immunoglobulin G (IgG) c) Immunoglobulin M (IgM) d) Ratio of IgM to IgG at both acute and convalescent serum samples.**
(TIF)

**S3 Fig. Relationship between estimated probability of a dengue virus infection (DENV) found during model testing for Model 9 (incorporating paired IgG, IgM, GM HAI, and acute GM HAI serological data) and the number of serological assays (RT-PCR, HAI, and EIA) that interpret the individual as having an infection.**
(TIF)

**S4 Fig. Acute and convalescent samples from cohort study data for serological assays, colored by whether they are predicted as infections using Model 9.** Green points represent no predicted infection while samples classified as an infection are split into two groups, those with no positive serological interpretations (RT-PCR, HAI, and EIA) in red while those with at least one positive serological interpretation are blue. a) Geometric mean titers of a haemagglutination inhibition assay (GM HAI) for all four serotypes of dengue virus. b) Immunoglobulin G (IgG) c) Immunoglobulin M (IgM) d) Ratio of IgM to IgG at both acute and convalescent serum samples.
(TIF)

**S5 Fig. Acute and convalescent data for serological assays, colored by probability of infection found during model testing for Model 9. a) Geometric mean titers of a haemagglutination inhibition assay (GM HAI) for all four**

serotypes of dengue virus. b) Immunoglobulin G (IgG) c) Immunoglobulin M (IgM) d) Ratio of IgM to IgG at both acute and convalescent sera samples.
(TIF)

**S6 Table. Comparison of accuracy metrics (sensitivity, specificity, and AUC ROC) of each model defined by what subset of data was used for training and testing.** For models with convergence issues an NA is presented.
(XLSX)

**S6 Fig. Comparison between simulation based estimates for boosts in GM HAI, IgG, and IgM that is experienced after an infection event as a function of acute GM HAI, IgG, and IgM titers individuals under the varying levels of observational noise.** The black solid lines represent the defined functional relationship between acute and convalescent titers after the boost and subsequent waning of titers between the samples.
(TIF)

**S7 Fig. (A) Estimated intercept for titer boost when acute titers are initially low for each assay.** (B) Estimated relationship between titer boost and acute titer for each assay.
(PNG)

## Acknowledgments

We are thankful for all efforts from the data collection team as well as the children and adults involved in the study.

## Author contributions

**Conceptualization:** Marco Hamins-Puertolas, Kathryn B. Anderson, Isabel Rodriguez-Barraquer.

**Data curation:** Darunee Buddhari, Alan L. Rothman, Kathryn B. Anderson.

**Formal analysis:** Marco Hamins-Puertolas, Kathryn B. Anderson, Isabel Rodriguez-Barraquer.

**Funding acquisition:** Darunee Buddhari, Stefan Fernandez, Aaron Farmer, Timothy Endy, Alan L. Rothman, Kathryn B. Anderson, Isabel Rodriguez-Barraquer.

**Investigation:** Marco Hamins-Puertolas, Darunee Buddhari, Taweewun Hunsawong, Anon Srikiatkhachorn.

**Methodology:** Marco Hamins-Puertolas, Henrik Salje, Angkana T. Huang, Derek A.T. Cummings, Adam Waickman, Stephen J. Thomas, Kathryn B. Anderson, Isabel Rodriguez-Barraquer.

**Project administration:** Darunee Buddhari, Stefan Fernandez, Aaron Farmer, Anon Srikiatkhachorn, Adam Waickman, Stephen J. Thomas, Timothy Endy, Alan L. Rothman, Kathryn B. Anderson.

**Resources:** Darunee Buddhari, Surachai Kaewhiran, Direk Khampaen, Sopon Iamsirithaworn.

**Software:** Marco Hamins-Puertolas.

**Supervision:** Darunee Buddhari, Stefan Fernandez, Aaron Farmer, Kathryn B. Anderson, Isabel Rodriguez-Barraquer.

**Validation:** Marco Hamins-Puertolas.

**Visualization:** Marco Hamins-Puertolas.

**Writing – original draft:** Marco Hamins-Puertolas, Kathryn B. Anderson, Isabel Rodriguez-Barraquer.

**Writing – review & editing:** Marco Hamins-Puertolas, Darunee Buddhari, Henrik Salje, Angkana T. Huang, Taweewun Hunsawong, Derek A.T. Cummings, Stefan Fernandez, Aaron Farmer, Surachai Kaewhiran, Direk Khampaen, Anon Srikiatkhachorn, Sopon Iamsirithaworn, Adam Waickman, Stephen J. Thomas, Timothy Endy, Alan L. Rothman, Kathryn B. Anderson, Isabel Rodriguez-Barraquer.

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
