## [Decision Letter · Decision Letter 0]

6 Jul 2025

PCOMPBIOL-D-25-00773

Linking multiple serological assays to infer dengue virus infections from paired samples using mixture models

PLOS Computational Biology

Dear Dr. Hamins-Puertolas,

Thank you for submitting your manuscript to PLOS Computational Biology. After careful consideration, we feel that it has merit but does not fully meet PLOS Computational Biology's publication criteria as it currently stands. Therefore, we invite you to submit a revised version of the manuscript that addresses the points raised during the review process.

Please submit your revised manuscript within 30 days Sep 05 2025 11:59PM. If you will need more time than this to complete your revisions, please reply to this message or contact the journal office at ploscompbiol@plos.org. Please include the following items when submitting your revised manuscript:

We look forward to receiving your revised manuscript.

Kind regards,

Moritz Kraemer, DPhil

Academic Editor

PLOS Computational Biology

Roger Kouyos

Section Editor

PLOS Computational Biology

**Additional Editor Comments:**

Thank you for submitting your work to PLOS Computational Biology. The reviewers recommended minor revisions -- see their comments attached to this email.

**Journal Requirements:**

At this stage, the following Authors/Authors require contributions: Marco Hamins-Puertolas, Darunee Buddhari, Henrik Salje, Angkana T. Huang, Taweewun Hunsawong, Derek A.T. Cummings, Stefan Fernandez, Aaron Farmer, Surachai Kaewhiran, Direk Khampaen, Anon Srikiatkhachorn, Sopon Iamsirithaworn, Adam Waickman, Stephen J. Thomas, Timothy Endy, Alan L. Rothman, Kathryn B. Anderson, and Isabel Rodriguez-Barraquer. Please ensure that the full contributions of each author are acknowledged in the "Add/Edit/Remove Authors" section of our submission form.

4) We do not publish any copyright or trademark symbols that usually accompany proprietary names, eg ©,  ®, or TM  (e.g. next to drug or reagent names). Therefore please remove all instances of trademark/copyright symbols throughout the text, including:

- TM on page: 31.

5) Thank you for including an Ethics Statement for your study. Please include:

i) A statement that formal consent was obtained (must state whether verbal/written) OR the reason consent was not obtained (e.g. anonymity). NOTE: If child participants, the statement must declare that formal consent was obtained from the parent/guardian.].

6) Please upload all main figures as separate Figure files in .tif or .eps format. For more information about how to convert and format your figure files please see our guidelines: 

7) We notice that your supplementary Figures, Tables, and information are included in the manuscript file. Please remove them and upload them with the file type 'Supporting Information'. Please ensure that each Supporting Information file has a legend listed in the manuscript after the references list.

8) Please ensure that the funders and grant numbers match between the Financial Disclosure field and the Funding Information tab in your submission form. Note that the funders must be provided in the same order in both places as well. Currently, "Military Infectious Disease Research Program (MIDRP)" is missing from the Funding Information tab.

9) Please provide a completed 'Competing Interests' statement, including any COIs declared by your co-authors. If you have no competing interests to declare, please state "The authors have declared that no competing interests exist". 

**Reviewers' comments:**

Reviewer's Responses to Questions

**Comments to the Authors:**

**Please note that one of the reviews is uploaded as an attachment.**

Reviewer #1: The review is uploaded as an attachment

Reviewer #2: The authors present a method to jointly analyse multiple serological assays from paired samples. They use mixture models, the application of which is layed out in the manuscript. They show that the use of additional assays improves the accuracy of infection detection. The manuscript is well written, the concepts are well explained, and the approach is well chosen and validated. I only have some minor comments that would help increase the clarity of the manuscript and future use of the methods proposed.

The authors discuss some extensions to their framework, including adding baseline titers and the time since symptom onset as covariates. It would be helpful for future applications if the authors would describe the methodology used in these additional steps more explicitly.

Similarly, the simulation of the synthetic data is performed using the serosim package. It would be helpful if, in addition to the parameter used (table S1) the model used for the generation of the data was given (in main of SM). In particular, it was not clear to me how the immune history was modelled.

Lastly, the model fitting approach is very briefly described. What algorithm was used here? This part would benefit from a bit more details.

**Have the authors made all data and (if applicable) computational code underlying the findings in their manuscript fully available?**

Reviewer #1: Yes

Reviewer #2: Yes

PLOS authors have the option to publish the peer review history of their article (what does this mean? ). If published, this will include your full peer review and any attached files.

**Do you want your identity to be public for this peer review?** For information about this choice, including consent withdrawal, please see our Privacy Policy .

Reviewer #1: **Yes: ** Victoria Cox

Reviewer #2: No

**Figure resubmission:**
---

## [Decision Letter · Decision Letter 1]

6 Nov 2025

Dear Dr Hamins-Puertolas,

We are pleased to inform you that your manuscript 'Linking multiple serological assays to infer dengue virus infections from paired samples using mixture models' has been provisionally accepted for publication in PLOS Computational Biology.

Best regards,

Roger Dimitri Kouyos

Section Editor

PLOS Computational Biology

Roger Kouyos

Section Editor

PLOS Computational Biology

Reviewer's Responses to Questions

**Comments to the Authors:**

Reviewer #1: The authors have revised the manuscript based on the suggestions made in the reviews. The clarity around the different models has been improved. The manuscript reads well and as I wrote in the original review, it is interesting and important work. Very minor spot: on line 230 in the tracked changes version, there is a double 'AA'.

Reviewer #2: All comments were well addressed.

**Have the authors made all data and (if applicable) computational code underlying the findings in their manuscript fully available?**

Reviewer #1: Yes

Reviewer #2: None

PLOS authors have the option to publish the peer review history of their article (what does this mean? ). If published, this will include your full peer review and any attached files.

**Do you want your identity to be public for this peer review?** For information about this choice, including consent withdrawal, please see our Privacy Policy .

Reviewer #1: **Yes: ** Victoria Cox

Reviewer #2: No

---

## [Editor Report · Acceptance letter]

PCOMPBIOL-D-25-00773R1

Linking multiple serological assays to infer dengue virus infections from paired samples using mixture models

Dear Dr Hamins-Puertolas,

I am pleased to inform you that your manuscript has been formally accepted for publication in PLOS Computational Biology. Your manuscript is now with our production department and you will be notified of the publication date in due course.

With kind regards,

Zsofia Freund
